# Compensating errors in inversions for subglacial bed roughness: same steady state, different dynamic response

Constantijn J. Berends[1], Roderik S. W. van de Wal[1,2], Tim van den Akker[1], William H. Lipscomb[3]

[1] Institute for Marine and Atmospheric research Utrecht, Utrecht University, Utrecht, The Netherlands
[2] Faculty of Geosciences, Department of Physical Geography, Utrecht University, Utrecht, The Netherlands
[3] Climate and Global Dynamics Laboratory, National Center for Atmospheric Research, Boulder, CO, USA

*Correspondence to*: Tijn Berends (c.j.berends@uu.nl)

**Abstract.** Subglacial bed roughness is one of the main factors controlling the rate of future Antarctic ice-sheet retreat, and also one of the most uncertain. A common technique to constrain the bed roughness using ice-sheet models is basal inversion, tuning the roughness to reproduce the observed present-day ice-sheet geometry and/or surface velocity. However, many other factors affecting ice-sheet evolution, such as the englacial temperature and viscosity, the surface and basal mass balance, and the subglacial topography, also contain substantial uncertainties. Using a basal inversion technique intrinsically causes any errors in these other quantities to lead to compensating errors in the inverted bed roughness. Using a set of idealised-geometry experiments, we quantify these compensating errors and investigate their effect on the dynamic response of the ice-sheet to a prescribed forcing. We find that relatively small errors in ice viscosity and subglacial topography require substantial compensating errors in the bed roughness in order to produce the same steady-state ice sheet, obscuring the realistic spatial variability in the bed roughness. When subjected to a retreat-inducing forcing, we find that these different parameter combinations, which per definition of the inversion procedure result in the same steady-state geometry, lead to a rate of ice volume loss that can differ by as much as a factor of two. This implies that ice-sheet models that use basal inversion to initialise their model state can still display a substantial model bias despite having an initial state which is close to the observations.

## 1 Introduction

One of the most worrying long-term consequences of anthropogenic climate change is sea-level rise due to mass loss of the Greenland and Antarctic ice sheets (Oppenheimer et al., 2019; Fox-Kemper et al., 2021). It is also one of the most uncertain consequences, with the projected sea-level contribution from the Antarctic ice sheet in 2100 under high-warming scenarios ranging from -2.5 cm (the minus sign indicating a sea-level drop) to 17 cm (Seroussi et al., 2020). Ice-dynamical processes are the main contributors to this uncertainty, which is demonstrated in the idealised (though extreme) ABUMIP experiment (Sun et al., 2020), which concerns instantaneous ice-shelf collapse under zero atmospheric or oceanic forcing, thereby eliminating uncertainties in the forcing. In this experiment, modelled sea-level rise differs by a factor of 10 among models, on time scales of a few centuries.

One of the main contributing factors to this ice-dynamical uncertainty is basal sliding, which is controlled by the conditions of the subglacial bed. Sun et al. (2020) showed that a substantial amount of the variance in the ABUMIP model ensemble could be explained by different assumptions about the relation between bed roughness, sliding velocity, and basal friction (the "sliding law"). These processes are difficult to constrain based on observational evidence; observations of the Antarctic

subglacial substrate are virtually non-existent, and direct observations of ice velocity are typically limited to the ice-sheet surface, which contains contributions from both basal sliding and vertical shearing. Since the latter is controlled by the ice viscosity, which too is very uncertain, disentangling the two terms is problematic.

An often-used approach for solving this problem is applying inversion techniques to estimate either the bed roughness or the

basal drag, by matching the observed ice thickness and/or surface velocity. Generally speaking, an inversion is a way to calculate the cause of an observed effect; since most physical problems instead consist of calculating the effect of an observed or postulated cause, this is called the "inverse problem". In the case of basal sliding, the forward problem consists of providing an ice-sheet model with a (spatially variable) value for bed roughness, and calculating the resulting ice-sheet geometry and/or velocity. The inverse problem consists of taking the (observed) geometry and/or velocity, and using that to invert for the bed

roughness. Different formulations of this approach exist, which differ in the observations the inversion aims to reproduce (e.g. ice-sheet geometry and/or velocity), in the quantity that is inverted for (bed roughness or basal drag), and in the mathematical techniques used to perform the inversion. A geometry-based approach was introduced by Pollard and DeConto (2012), and adapts the bed roughness during a forward simulation until the model reaches a steady-state ice geometry that matches the observations. The bed roughness is changed based on the local difference between the modelled and the observed ice thickness;

if the ice is too thick (thin), the bed roughness is decreased (increased), based on the idea that a lower (higher) bed roughness leads to increased (decreased) ice flow, and therefore thinning (thickening). This approach has since been adopted, with minor variations, in several ice sheet models, e.g. f.ETISh (Pattyn, 2017), PISM (Albrecht et al., 2020), and CISM (Lipscomb et al., 2021). The velocity-based approach is used in e.g. Elmer/ice (Gagliardini et al., 2013) and ISSM (Larour et al., 2012), and often inverts directly for basal drag, without making any assumptions about the sliding law. In this approach, the model is not

run forward in time; instead, the basal drag field is iteratively adapted until the modelled velocity field for the observed geometry matches the observed velocity. Typically, more elaborate mathematical techniques are used to update the inverted field than in the geometry-based approach. For example, the drag may be computed by defining and iteratively minimising a cost function that represents the mismatch between the modelled and observed velocity (e.g. Arthern and Gudmundsson, 2010; Gagliardini et al., 2013; Arthern et al., 2015). The cost function typically includes a term quantifying unwanted small-

wavelength terms in the solution, which can arise as a result of overfitting. Since the velocity-based approach does not make any assumptions about the dynamic (steady) state of the geometry, it generally leads to a more pronounced model drift compared to the geometry-based approach in forward experiments (Seroussi et al., 2019).

These inversion approaches share the underlying assumption that all ice-sheet properties other than the bed roughness are known accurately enough for such an inversion to be meaningful, i.e. that any differences between the modelled and the observed ice-sheet state are mostly due to errors in the modelled bed roughness, and that those errors can be corrected by applying an inversion. This means that, due to the nature of the inversion procedure, any modelled errors in the other ice-sheet properties will lead to compensating errors in the inverted bed roughness. For example, if the modelled ice viscosity overestimates the real value, then the modelled ice velocities due to viscous deformation will be too low, and the modelled steady-state ice sheet will be too thick. The inversion procedure will compensate for this mismatch by lowering the bed roughness, increasing the sliding velocities (and thinning the ice, in the case of geometry-based inversion methods) until the modelled ice sheet once again matches the observed state. This implies that the result of a basal inversion will contain not just (an approximation of) the realistic bed roughness, but also the sum of compensating errors that arise from modelled errors in other ice-sheet quantities.

Several studies have already investigated these compensating errors in different settings. Seroussi et al. (2013) studied the effect of uncertainties in the thermal regime of the Greenland ice sheet on the inverted bed roughness, and on future projections of ice-sheet volume. They found that, while the effect on the inverted bed roughness was substantial, the differences in projected ice volume change were minimal. Perego et al. (2014) studied the effect of uncertainties in surface mass balance and ice thickness on inversions of bed roughness for the Greenland ice sheet. They presented a method that could simultaneously invert for surface mass balance, basal topography, and basal roughness, thus providing a better fit to the observed velocity and a more stable ice sheet. Babaniyi et al. (2021) studied the effect of errors in the modelled ice rheology on the inverted bed roughness in an idealised setting. They found that uncertainties in the rheology and viscosity of the ice could lead to significant biases in the inverted roughness. Arthern et al. (2015) and Ranganathan et al. (2021) presented methods for simultaneously inverting for both viscosity and basal slipperiness. These methods provide accurate estimates of both velocity and ice thickness, as long as uncertainties in the observed ice thickness and bed topography are small (Ranganathan et al., 2021).

In this study, we investigate the compensating errors in a geometry+velocity-based inversion approach, and how they affect the uncertainty in projections of ice-sheet retreat. As a modelling tool we use the vertically-integrated ice-sheet model IMAU-ICE (Berends et al., 2022), which we describe briefly in Sect. 2.1. In Sect. 2.2 we present a novel variation on the geometry-based inversion approach, which uses a flowline-averaged anomaly method to adapt the bed roughness field. We apply this model set-up to two idealised-geometry ice sheets, which we describe in Sect. 3. In Sect. 4.1 we demonstrate that our novel inversion procedure can reproduce the known bed roughness in settings with freely moving ice margins and/or grounding lines. In Sect. 4.2 we present a series of experiments where we introduce errors in other ice-sheet model components before performing the inversion, which results in an erroneous inverted bed roughness, even though, as a construct of the inversion procedure, the resulting steady-state ice sheet is similar. In Sect. 4.3 we investigate the effect of these compensating errors on the dynamic response of the ice-sheet to a schematic retreat-inducing forcing. We show that, even though the respective errors

in the bed roughness and the other model components compensate each other in terms of steady-state ice-sheet geometry, this is not necessarily the case for the dynamic response. We quantify the difference in ice-sheet models with nearly identical steady-state geometries in their rate of sea-level contribution under a forced retreat as a result of the compensating errors. We discuss the implications of these findings in Sect. 5.

## 2 Methodology

### 2.1 Ice-sheet model

IMAU-ICE is a vertically-integrated ice-sheet model, which has been specifically designed for large-scale, long-term simulations of ice-sheet evolution (Berends et al., 2022). It solves the depth-integrated viscosity approximation (DIVA; Goldberg, 2011; Lipscomb et al., 2019) to the stress balance, which is similar to the hybrid SIA/SSA, but which remains close to the full-Stokes solution at significantly higher aspect ratios (Berends et al., 2022). Proper grounding-line migration is achieved by using a sub-grid friction-scaling scheme, based on the approaches used in PISM (Feldmann et al., 2014) and CISM (Leguy et al., 2021).

For this study, a new sliding law was added to IMAU-ICE, based on the work of Zoet and Iverson (2020). This recent work presents a sliding law based on laboratory experiments, contrasting with previous sliding laws which were based chiefly on theoretical considerations. Here, the basal shear stress $\tau_b$ depends on the basal velocity $u_b$ as follows:

$$\tau_b = \widehat{u}_b N \tan \varphi \left( \frac{|u_b|}{|u_b| + u_0} \right)^{1/p}.$$

(1)

Here, $N$ is the (effective) overburden pressure, which we assume to be identical to the ice overburden pressure (i.e. no subglacial water), $\widehat{u}_b$ is the unit vector parallel to the basal velocity, and $\varphi$ is the bed roughness, expressed as a till friction angle. By default, the exponent $p$ has a value of $p = 3$, and the transition velocity $u_0$ has a value of $u_0 = 200 \text{ m yr}^{-1}$. At low sliding velocities, this sliding law behaves like a Weertman-type power law (Weertman, 1957), with the basal shear stress approaching zero as the basal velocity approaches zero. At high sliding velocities, the basal shear stress asymptotes to the Coulomb friction limit (Iverson et al., 1998). This two-regime behaviour agrees with the theoretical considerations underlying previous sliding laws (e.g. Schoof, 2005; Tsai et al., 2015).

### 2.2 Inversion procedure

For this study, we developed a novel inversion procedure. It is based on the procedure used in CISM (Lipscomb et al., 2021), which in turn is a variation on the geometry-based approach from Pollard and DeConto (2012). In the CISM procedure, as in the Pollard and DeConto approach, the ice-sheet model is run forward in time, and the bed roughness field is adapted based

on the difference between the modelled and the target ice-sheet. However, whereas the Pollard and DeConto approach only considers the mismatch in ice thickness, a newer, unpublished approach in CISM additionally includes the mismatch in surface velocity, leading to faster convergence (since the velocity responds more quickly to changes in bed roughness than the geometry). We extend this approach by adopting a flowline-averaged, rather than a purely local scheme to calculate the mismatch in terms of ice thickness and velocity. The rationale behind this is that changing the bed roughness at any location will affect the ice geometry and velocity not just at that location, but also upstream and downstream. Reducing the basal roughness at one location will increase the ice velocity along the entire flowline, causing the ice both locally and upstream to become thinner. By including these effects in the inversion procedure, numerical stability is improved, and artefacts arising from differences in the flotation mask between the modelled and the target state are reduced. The bed roughness produced by the inversion is not affected by these changes, as the inclusion of a regularisation term usually ensures that the bed roughness converges to the same solution. The approach outlined here mainly improves the numerical stability and robustness under changing ice sheet/ice shelf/ocean masks of the inversion. This is shown in Appendix A, where we compare the convergence behaviour of our new inversion procedure to a method currently used in CISM, which also uses both the geometry and velocity mismatch, but without the flowline-averaging approach.

Let $\boldsymbol{p} = [x, y]$ be a point on the ice sheet. We divide the flowline passing through $\boldsymbol{p}$ into an upstream part $\boldsymbol{L}_u(\boldsymbol{p}, s)$ and a downstream part $\boldsymbol{L}_d(\boldsymbol{p}, s)$, which can be found by integrating the ice surface velocity field $\widehat{\boldsymbol{u}} = \frac{\boldsymbol{u}}{|\boldsymbol{u}|}$:

$$\boldsymbol{L}_u(\boldsymbol{p}, s + ds) = \boldsymbol{L}_u(\boldsymbol{p}, s) - \widehat{\boldsymbol{u}}\big(\boldsymbol{L}_u(\boldsymbol{p}, s)\big)ds, \tag{2a}$$

$$\boldsymbol{L}_d(\boldsymbol{p}, s + ds) = \boldsymbol{L}_d(\boldsymbol{p}, s) + \widehat{\boldsymbol{u}}\big(\boldsymbol{L}_d(\boldsymbol{p}, s)\big)ds, \tag{2b}$$

$$\boldsymbol{L}_u(\boldsymbol{p}, 0) = \boldsymbol{L}_d(\boldsymbol{p}, 0) = \boldsymbol{p}. \tag{2c}$$

Here, $s$ is the distance along the flowline. In the upstream (downstream) direction, the integral is terminated at $s_u$ ($s_d$) at the ice divide (ice margin), i.e. when $\boldsymbol{u} = \boldsymbol{0}$ ($H = 0$), so that:

$$\boldsymbol{u}\left(\boldsymbol{L}_u(\boldsymbol{p}, s_u(\boldsymbol{p}))\right) = \boldsymbol{0}, \tag{3a}$$

$$H\left(\boldsymbol{L}_d(\boldsymbol{p}, s_d(\boldsymbol{p}))\right) = 0. \tag{3b}$$

In order to calculate the rate of change $\frac{d\varphi}{dt}$ of the till friction angle $\varphi$, the velocity mismatch (defined as the difference between the modelled absolute surface velocity $|\boldsymbol{u}_m|$ and the target absolute surface velocity $|\boldsymbol{u}_t|$) is averaged over both the upstream (Eq. 4a) and downstream (Eq. 4b) part of the flowline, whereas the ice thickness mismatch is evaluated only in the upstream

direction (Eq. 4c; preliminary experiments showed that including a downstream ice thickness term was detrimental to the results):

$$I_1(\boldsymbol{p}) = \int_{s=0}^{s_u(\boldsymbol{p})} \left( \frac{|\boldsymbol{u}_m(\boldsymbol{L}_u(\boldsymbol{p},s))| - |\boldsymbol{u}_t(\boldsymbol{L}_u(\boldsymbol{p},s))|}{u_0} \right) w_u\big(s, s_u(\boldsymbol{p})\big) ds, \tag{4a}$$

$$I_2(\boldsymbol{p}) = \int_{s=0}^{s_d(\boldsymbol{p})} \left( \frac{|\boldsymbol{u}_m(\boldsymbol{L}_d(\boldsymbol{p},s))| - |\boldsymbol{u}_t(\boldsymbol{L}_d(\boldsymbol{p},s))|}{u_0} \right) w_d\big(s, s_d(\boldsymbol{p})\big) ds, \tag{4b}$$

$$I_3(\boldsymbol{p}) = \int_{s=0}^{s_u(\boldsymbol{p})} \left( \frac{H_m(\boldsymbol{L}_u(\boldsymbol{p},s)) - H_t(\boldsymbol{L}_u(\boldsymbol{p},s))}{H_0} \right) w_u\big(s, s_u(\boldsymbol{p})\big) ds. \tag{4c}$$

Here, $I_1$ represents the distance-weighted average of the velocity anomaly over the half-flowline upstream of $\boldsymbol{p}$; $I_2$ represents the distance-weighted average of the velocity anomaly over the half-flowline downstream of $\boldsymbol{p}$; and $I_3$ represents the distance-weighted average of the geometry anomaly over the half-flowline upstream of $\boldsymbol{p}$. The default values for the scaling parameters are $u_0 = 250$ m yr$^{-1}$, $H_0 = 100$ m. The linear scaling functions $w_u, w_d$ serve to assign more weight to anomalies close to $\boldsymbol{p}$, decreasing to zero at the ends of the flowline, as well as to normalise the integral:

$$w_u\big(s, s_u(\boldsymbol{p})\big) = \frac{2}{s_u(\boldsymbol{p})} \left( 1 - \frac{s}{s_u(\boldsymbol{p})} \right), \tag{5a}$$

$$w_d\big(s, s_d(\boldsymbol{p})\big) = \frac{2}{s_d(\boldsymbol{p})} \left( 1 - \frac{s}{s_d(\boldsymbol{p})} \right). \tag{5b}$$

The scaling functions are constructed such that $\int_{s=0}^{s=s_u(\boldsymbol{p})} w_u ds = \int_{s=0}^{s=s_d(\boldsymbol{p})} w_d ds = 1$. It is possible that integrating a finite
distance from $\boldsymbol{p}$, rather than over the entire flowline, might improve the rate of convergence; we did not perform any preliminary experiments to test this. The three line integrals from Eqs. 4a-c are then added together, and scaled with the local ice thickness $H(\boldsymbol{p})$ and velocity $|\boldsymbol{u}(\boldsymbol{p})|$. This reflects the fact that bed roughness underneath slow-moving and/or thin ice has less effect on the large-scale ice-sheet geometry than the roughness underneath fast-flowing and/or thick ice:

$\quad I_{\text{tot}}(\boldsymbol{p}) = \big( I_1(\boldsymbol{p}) + I_2(\boldsymbol{p}) + I_3(\boldsymbol{p}) \big) R(\boldsymbol{p}),$  (6)

$\quad R(\boldsymbol{p}) = \frac{|\boldsymbol{u}(\boldsymbol{p})| H(\boldsymbol{p})}{u_s H_s}, 0 \leq R(\boldsymbol{p}) \leq 1.$  (7)

By default, the scaling parameters are $u_s = 3{,}000$ m yr$^{-1}$, $H_s = 300$ m. These values are based on preliminary experiments to attain fast convergence without creating numerical artefacts. Finally, the rate of change $\frac{d\varphi}{dt}$ of the till friction angle $\varphi$ can be
calculated:

$$\frac{d\varphi(\boldsymbol{p})}{dt} = -\frac{\varphi(\boldsymbol{p}) I_{\text{tot}}(\boldsymbol{p})}{t_s}. \tag{8}$$

The default value for the time scale is $t_s = 10$ yr, again based on preliminary experiments to balance the convergence rate against the numerical stability of the procedure. While the flowline integrals in Eqs. 4a-c are calculated over the entire flowline (including floating ice), $\frac{d\varphi}{dt}$ is calculated only for grounded ice; it is then extrapolated to fill the entire model domain using a simple Gaussian kernel. This approach helps to prevent artefacts in grid cells that switch over time between grounded and floating, or ice-covered and ice-free states, which typically present as individual or clustered grid cells where the iterative roughness adjustment overshoots, quickly diverging to extreme values.

The routine performing these calculations is run asynchronously from the other components of the ice-sheet model, with a time step of $\Delta t_\varphi = 5$ yr. The till friction angle is updated every time this routine is called:

$$\varphi_{n+1} = F_2\left(\varphi_n + \Delta t_\varphi F_1\left(\frac{d\varphi}{dt}\right)\right). \tag{9}$$

Here, $F_1$ and $F_2$ are Gaussian smoothing filters, with their respective radii defined relative to the grid resolution: $\sigma_1 = \frac{\Delta x}{1.5}$, $\sigma_2 = \frac{\Delta x}{4}$. These filters serve as a regularisation of the bed roughness, to prevent overfitting. Pattyn (2017) uses a similar regularisation approach, with a Savitsky-Golay filter instead of a Gaussian filter. Pollard and DeConto (2012) do not report any regularisation term in their inversion, while in CISM, the inclusion of a dH/dt term likely results in some smoothing. The radii of the two Gaussian filters, which were determined during preliminary experiments, are the lowest values we found that effectively repress small-wavelength terms in the inverted bed roughness, which are most likely a result of overfitting (Habermann et al., 2012). Increasing the radii of the filters does not significantly affect the inverted roughness until it is increased to several grid cells. Roughness variations of a small spatial scale could therefore potentially be obscured by the smoothing in our approach. However, such small variations would quickly approach the ice-dynamical limit of roughness variations that can be resolved by inverting from surface observations (about 50 ice thicknesses; Gudmundsson and Raymond, 2008), so this would likely not pose a serious problem in practical applications. The degree of overfitting in our approach is explored in more detail in Appendix A, where we demonstrate that it does not pose a significant problem.

Our inversion method does not include weighting of the velocity/elevation mismatch based on uncertainty estimations in the observations. However, including these weights in the method would not be difficult, and is worth considering when applying this method to the Greenland and/or Antarctic ice sheets.

It might be possible to improve upon the inversion procedure presented here, achieving faster or more robust convergence, or better computational performance. For example, our flowline-averaged approach might be difficult to implement in parallel

models with a distributed-memory architecture (i.e., where a processor might not have access to all the data on a flowline), which is not the case in IMAU-ICE. However, the aim of this manuscript is not to find the most efficient way to perform a basal inversion, but rather to investigate the uncertainties that remain in the result of that inversion even when the procedure itself works perfectly.

## 2.3 Perfect-model approach

In order to quantify the compensating errors from one particular model component, we use what we call a perfect-model approach. We first use the ice-sheet model to calculate the steady-state ice-sheet geometry for a known bed roughness field in a simulation we call the "target run". The known bed roughness will be called the target roughness, and the resulting ice-sheet the target geometry. If we then apply the inversion routine, with all model parameters set to the same values as were used to create the target geometry, then theoretically the resulting inverted bed roughness (which we call the unperturbed roughness)

should be exactly the same as the target roughness. The difference between the unperturbed roughness and the target roughness is the model error of the inversion routine. If the inversion procedure works adequately, this error should be small.

We then perform a "perturbed" inversion, where we change one or more of the model parameters/components (e.g. viscosity, SMB, subglacial topography) with respect to the target run. As long as the change is small enough that its effect on the steady-state geometry can be compensated for by a change in bed roughness, the inversion will produce an ice sheet that still matches

the target geometry and velocity, but with a different bed roughness, which we call the perturbed roughness. The difference between the perturbed and unperturbed roughness is the compensating error in the bed roughness caused by the error in the model parameter that was changed in the perturbed run. This procedure is illustrated schematically in Fig. 1.

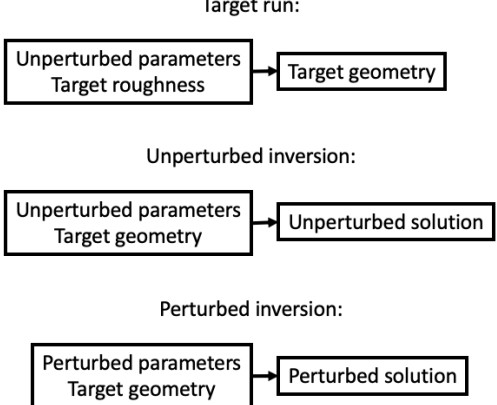

**Figure 1: Schematic illustration of the perfect-model approach used in this study.**

## 3 Idealised-geometry ice sheets

### 3.1 Experiment I: radially symmetrical ice sheet

The first of our two idealised-geometry ice sheets is based on the EISMINT-1 "moving margin" experiment (Huybrechts et al., 1996). It describes an ice-sheet on an infinite, non-deformable flat bed, with a radially symmetrical surface mass balance which is independent of the ice-sheet geometry:

$$M(r) = \min\left(M_{\max}, S(E - r)\right). \tag{10}$$

The values of the parameters are listed in Table 1; the radial distance $r$ from the grid centre is expressed in metres. The ice viscosity is described by a uniform value of Glen's flow law factor $A$ (i.e. no thermomechanical coupling). Lastly, we introduce a non-uniform till friction angle:

$$\varphi(x,y) = \varphi_{\max} - (\varphi_{\max} - \varphi_{\min})e^{\frac{-1}{2}\left(\left(\frac{x-x_c}{\sigma_x}\right)^2 + \left(\frac{y-y_c}{\sigma_y}\right)^2\right)}. \tag{11}$$

The values of the parameters listed in Table 1.

**Table 1: Parameter values for experiment I.**

| Parameter | Value | Description |
|---|---|---|
| $M_{\max}$ | 0.5 m yr$^{-1}$ | Maximum accumulation rate |
| $E$ | 400 km | Radius of accumulation zone |
| $S$ | $10^{-5}$ yr$^{-1}$ | Melt rate increase over radial distance from grid centre |
| $A$ | $10^{-16}$ Pa$^{-3}$ yr$^{-1}$ | Glen's flow law factor |
| $\varphi_{\min}$ | 0.1° | Till friction angle in the centre of the ice stream |
| $\varphi_{\max}$ | 5° | Till friction angle outside the ice stream |
| $x_c$ | 0 m | x-coordinate of ice-stream centre |
| $y_c$ | -400 km | y-coordinate of ice-stream centre |
| $\sigma_x$ | 50 km | x-direction ice-stream half-width |
| $\sigma_y$ | 300 km | y-direction ice-stream half-width |

The equation thus describes a strip of reduced bed roughness running along the negative y-axis of the domain, which results in the formation of an ice stream with higher ice velocities, and a protruding ice lobe, as illustrated in Fig. 2. The ice sheet is initialised to a steady state by integrating the model through time for 50,000 years.

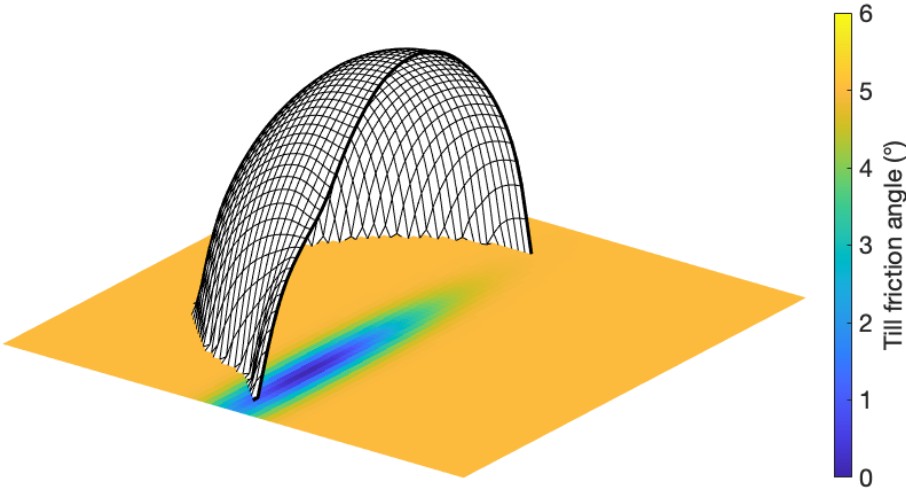

**Figure 2: Bed roughness and steady-state ice-sheet geometry in the EISMINT-based experiment I. Black lines on the ice surface are just for illustration. They do not correspond to the model grid.**

### 3.2 Experiment II: laterally symmetrical ice stream with shelf

The second idealised-geometry ice sheet is based on the MISMIP+ geometry (Asay-Davis et al., 2016). This describes a laterally symmetric glacial valley, about 800 km long and 80 km wide, with a slightly over-deepening bed, followed by a sill, before dropping sharply into a deep ocean. A uniform accumulation rate of 0.3 m yr$^{-1}$ leads to the formation of a fast-flowing ice stream feeding into a small embayed shelf. The grounding line rests on a retrograde slope, kept in place by buttressing forces. As in experiment I, we introduce a non-uniform bed roughness, which is again described by Eq. 11; the parameters for

this experiment are listed in Table 2. Following the MISMIP+ protocol set out by Asay-Davis et al. (2016), the uniform value for Glen's flow law factor $A = 1.13928 \cdot 10^{-17}$ Pa$^{-3}$yr$^{-1}$ is tuned to achieve a steady-state geometry with a mid-stream grounding-line position at $x = 450$ km, in the middle of the retrograde-sloping part of the bed. The resulting ice-sheet geometry is illustrated in Fig. 3. The ice sheet is initialised to a steady state by integrating the model through time for 50,000 years.

**Table 2: Parameter values for experiment II.**

| Parameter | Value | Description |
|---|---|---|
| $\varphi_{\min}$ | 1° | Till friction angle in the centre of the ice stream |
| $\varphi_{\max}$ | 5° | Till friction angle outside the ice stream |
| $x_c$ | -50 km | x-coordinate of ice-stream centre |
| $y_c$ | 0 km | y-coordinate of ice-stream centre |
| $\sigma_x$ | 150 km | x-direction ice-stream half-width |
| $\sigma_y$ | 15 km | y-direction ice-stream half-width |

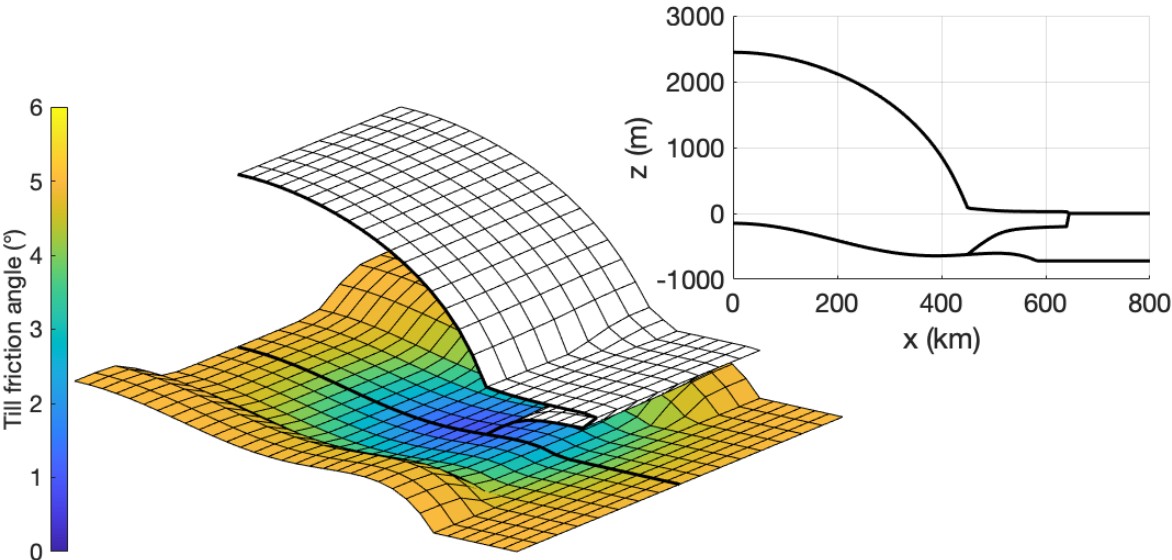

**Figure 3: Bed roughness and steady-state ice-sheet geometry in the MISMIP+-based experiment II. Black lines on the ice surface are only for illustration. They do not correspond to the model grid. The top-right panel shows a transect at y = 0, along the central flowline.**

## 4 Results

### 4.1 Unperturbed inversions

In order to verify that the inversion procedure is working properly, we first apply it to both idealised-geometry experiments with all model parameters unchanged. For experiment I, we perform these unperturbed inversions at resolutions of 40, 20, and 10 km; for experiment II we use values of 5 km and 2 km. The 50,000-yr steady-state initialisation is performed separately at all resolutions. The till friction angle is initialised with a uniform value of $\varphi = 5°$, and the model is run forward in time for 100,000 yr. With this choice of initial value, the bed roughness typically converges to a stable solution within ~30,000 years (as demonstrated by the additional experiments in Appendix A).

The resulting inverted bed roughness fields for both sets of simulations are shown in Figs. 4 and 5, respectively. The errors in the inverted bed roughness, and the resulting ice-sheet geometry and velocity, are very small at all resolutions and in both experiments (typically < 5% for the bed roughness, < 5 m for the surface elevation, and < 5 % for the surface velocity), indicating that the inversion procedure works well in the simple geometries of these two experiments.

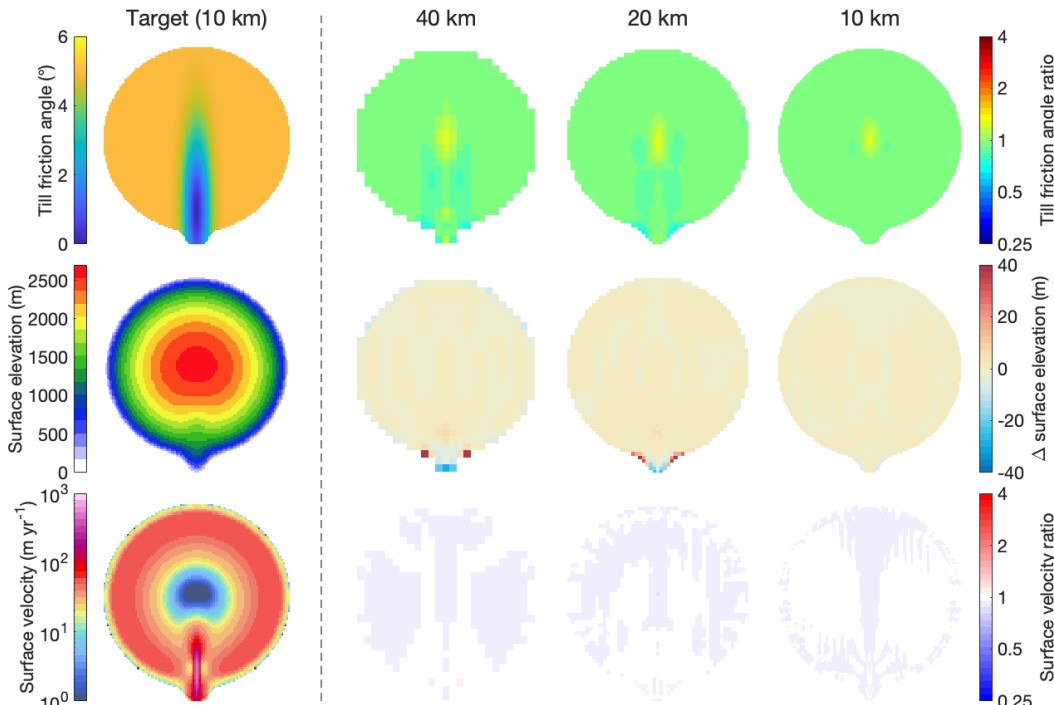

**Figure 4: Unperturbed inverted bed roughness, surface elevation, and surface velocity in experiment I at different resolutions, compared to the target. Top row: till friction angle; middle row: surface elevation; bottom row: surface velocity. For the target run (first column), absolute values are shown (colour scales on the left); for the three unperturbed inversions (second – fourth columns), errors with respect to the target are shown (colour scales on the right). For the till friction angle and the surface velocity, the ratios between the inverted and the target values are shown, using a logarithmic colour scale.**


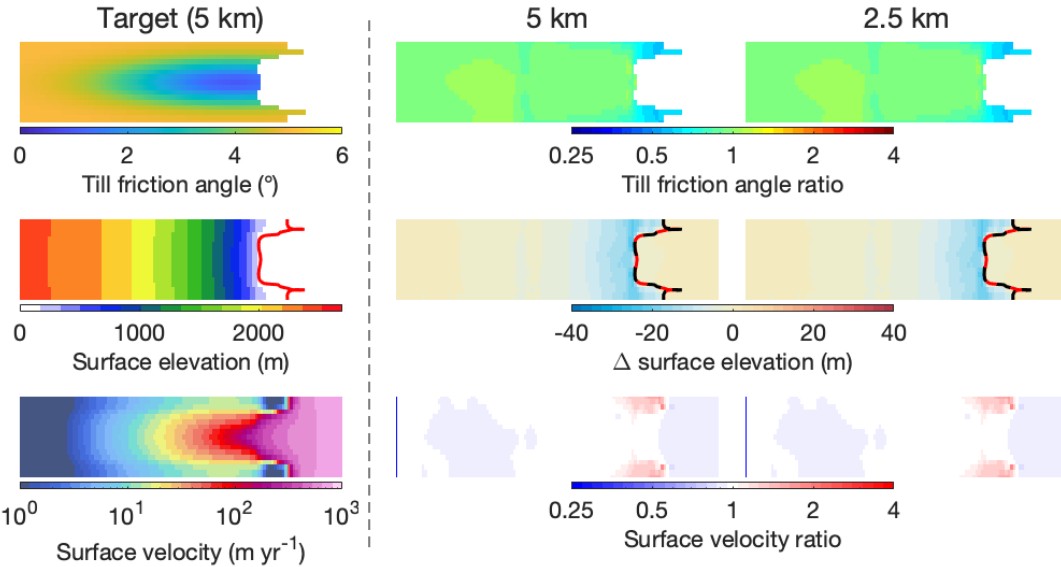

**Figure 5: Unperturbed inverted bed roughness, surface elevation, and surface velocity in experiment II at different resolutions, compared to the target. Top row: till friction angle; middle row: surface elevation; bottom row: surface velocity. For the target run (first column), absolute values are shown; for the three unperturbed inversions (second and third columns), errors with respect to the target are shown. The grounding line in the target (inverted) geometry is indicated by a solid red (dashed black) line.**

## 4.2 Perturbed inversions

To quantify the compensating errors in the inverted bed roughness, we perform a number of perturbed inversions, where we introduce errors in several model components. First, we increase (decrease) the uniform Glen's flow law factor A by a factor of 1.25. We assume that, in reality, this factor depends on the englacial temperature through an Arrhenius relation. The uncertainty in the annual mean surface temperature during the last glacial cycle is about 1 K for Antarctica (Jouzel et al., 2007) and 4 K for Greenland (Alley et al., 2000; Kindler et al., 2014). In realistic applications, a flow enhancement factor is often

applied to account for anisotropic rheology and damage. Since estimated values of this factor differ significantly (Ma et al., 2010), an uncertainty of an order of magnitude is plausible, but we chose a smaller range to ensure that the inversion procedure was still able to reproduce the target geometry. Second, we increase (decrease) the SMB by a factor of 1.05. This seemingly small range is motivated by the fact that, for simplicity's sake, we alter the SMB over the entire model domain. Whereas estimates of local mass balance contain significant uncertainties, ice-sheet-integrated values are additionally constrained by

satellite gravimetry, so that an uncertainty of 5% seems plausible (Fettweis et al., 2020). Next, we increase (decrease) the transition velocity $u_0$ in the Zoet-Iverson sliding law by a factor of 2, and we increase (decrease) the exponent $p$ in the sliding law by 2. Zoet and Iverson (2020) report a range of transition velocities between 50 and 200 m/yr, whereas in CISM a default value of 200 m/yr is used. For the exponent, Zoet and Iverson (2020) report a value of 5, CISM uses a value of 3, and a value of 1 yields a linear sliding law, which is still used in some ice-sheet models. We also perform two perturbed inversions where

we add an error to the bed topography of ±10 % of the ice thickness, resulting in a bump (depression) of just over 250 m beneath the ice divide. The ice thickness is adjusted accordingly to keep the surface elevation unchanged. While the surface elevation of the Greenland and Antarctic ice sheets is generally known very accurately, estimates of ice thickness and bedrock elevation are based on interpolation of local radar measurements. In the BedMachine Greenland v4 dataset (Morlighem et al, 2017), the reported uncertainty in the bedrock elevation exceeds 10% of the ice thickness over about 30% of the ice sheet. Our

choice of increasing/decreasing the estimated ice thickness by 10% everywhere therefore serves as an upper bound, as it is unlikely that all of the data and extrapolations are biased in the same direction.

These five parameters (viscosity, SMB, transition velocity, exponent, topography), each with a high and a low value, result in 10 perturbed inversion simulations. The resulting errors in the inverted bed roughness, steady-state ice geometry, and surface

velocity for experiment I are shown in Fig. 6.

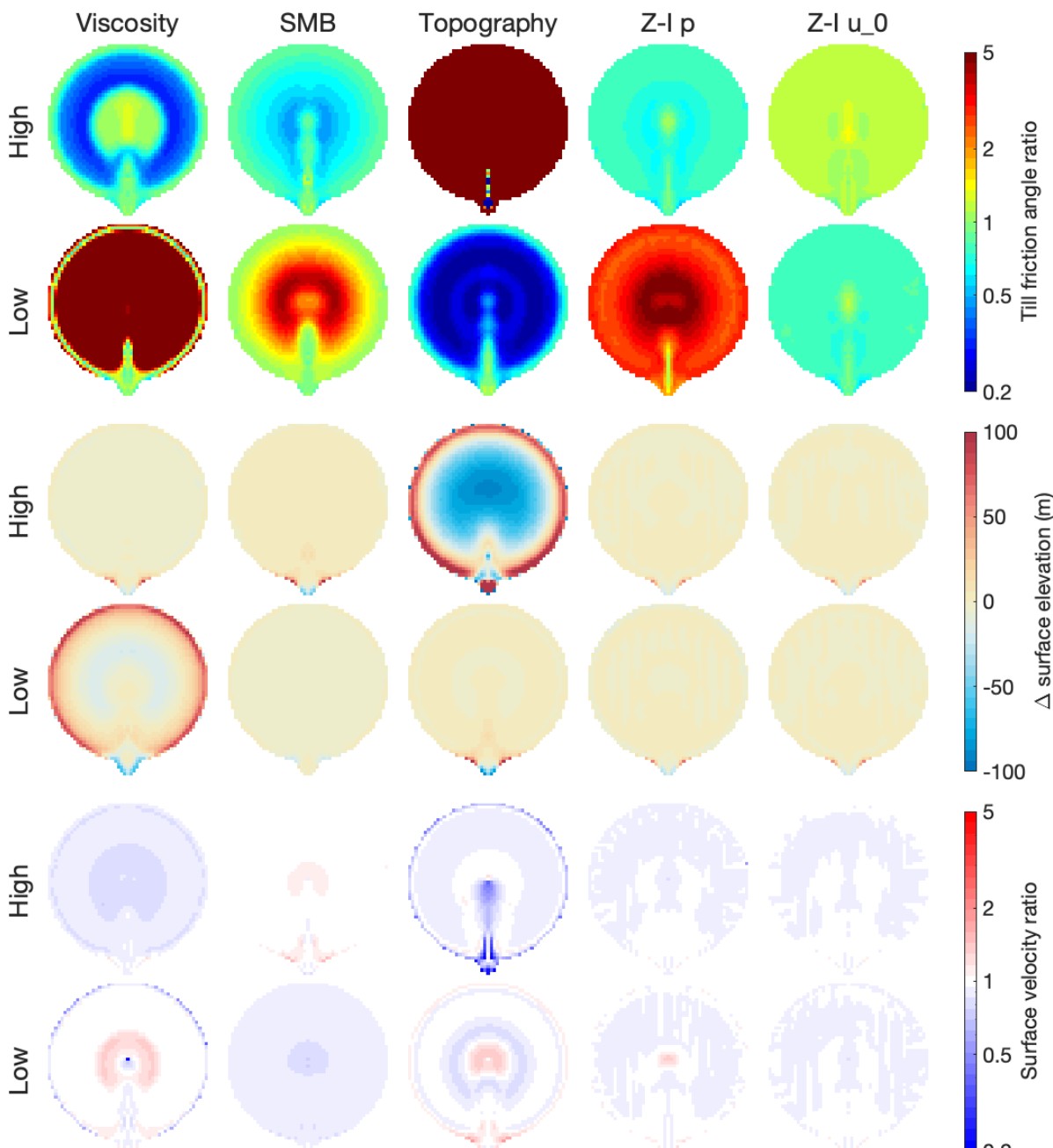

**Figure 6: Errors in inverted bed roughness, surface elevation, and surface velocity (relative to the target) for the perturbed inversions of experiment I. The top two rows show the errors in the bed roughness for the high and low perturbed inversions; the middle two rows show the errors in the steady-state surface elevation; and the bottom two rows show the errors in the surface velocity. Each column represents a single perturbed model parameter: viscosity (i.e., Glen's flow law factor A), surface mass balance, subglacial topography, and the exponent $p$ and transition velocity $u_0$ in the Zoet-Iverson sliding law.**


The top-leftmost panel in Fig. 6 shows the error in the inverted bed roughness for the high-viscosity perturbed inversion. In this experiment, the overestimated ice viscosity means that the ice flow due to vertical shearing is underestimated, which is compensated for by decreasing the bed roughness, leading to increased basal sliding. The leftmost panels in the third and fifth rows of Fig. 6 show the errors in the resulting steady-state ice geometry and surface velocity, which are negligibly small. For these two quantities, the errors in the viscosity and the bed roughness are indeed compensating errors. This is true for almost all perturbed inversions, except for the low-viscosity and high-topography runs (high-topography means an added depression in the bedrock, such that the target ice thickness is overestimated). In these two experiments, the added perturbations cause the deformational ice flow to be overestimated so much that even preventing all basal sliding cannot entirely compensate for this perturbation. Note that this results from perturbing Glen's flow law factor $A$ by a factor of 1.25, which is rather conservative. In realistic applications, the uncertainty in this quantity is typically an order of magnitude.

The underestimated value of the Zoet-Iverson sliding law exponent $p = 1$ (Fig. 6, fourth column, lower set of rows), which implies a linear sliding law, yields negligible errors in the geometry and velocity, but results in the inverted bed roughness being overestimated by a factor of 3 on average. The overestimated value of $p = 5$ yields negligible differences, as do both over- and underestimated values of the transition velocity $u_0$.

In the remaining four perturbed viscosity / mass balance / topography simulations, the errors in the inverted geometry are acceptably small, compared to the errors reported for initialised models in realistic intercomparison projects (e.g., initMIP-Greenland; Goelzer et al., 2018). The errors in the inverted bed roughness, however, are as large or larger than the "signal" of the prescribed bed roughness pattern (i.e. ~5° of till friction angle change in the ice-stream area). These errors show prominent spatial patterns, despite the fact that the perturbations are spatially uniform. This implies that one should be cautious when interpreting the spatial patterns yielded by a basal inversion procedure, as they could reflect errors in some other physical quantity rather than realistic variations in bed roughness.

For experiment II, we perform the same set of perturbed inversions as for experiment I, introducing the same perturbations to the ice viscosity, the surface mass balance, the subglacial topography, and the sliding law parameters. We additionally perturb the sub-shelf melt rate, applying values of $\pm 1$ m/yr (in the target run, no basal melt is applied). The results of the perturbed inversions are shown in Fig. 7. The results of the perturbed Zoet-Iverson sliding law transition velocity $u_0$ are omitted, since that has only a small effect. Similar to experiment I, the relatively small errors introduced in the ice viscosity, mass balance, and subglacial topography lead to large errors in the inverted bed roughness, but still produce a steady-state ice geometry that is close to the target geometry. The only exceptions are, again, the low-viscosity and high-topography runs, as well as the low-BMB run (i.e., too much sub-shelf melt), where the ice flow is increased more than can be compensated for by increasing the basal friction. However, even here the errors in the inverted geometry are relatively small. The errors in the inverted velocities are mostly small, except for the inversions with the perturbed sub-shelf melt rates. While these inversions produce relatively

accurate geometries (about 120 m of ice loss near the grounding line in the increased-melt simulations), they contain large errors in the shelf velocities (about -500 m/yr in the increased-melt simulation, relative to a target value of about 1,000 m/yr).

As in experiment I, the introduced perturbations (which are spatially uniform) lead to prominent spatial patterns in the inverted
bed roughness, with the errors being as large as the actual (prescribed) signal. This underlines the conclusion that spatial patterns in inverted bed roughness do not necessarily correspond to spatial patterns in the true bed roughness.

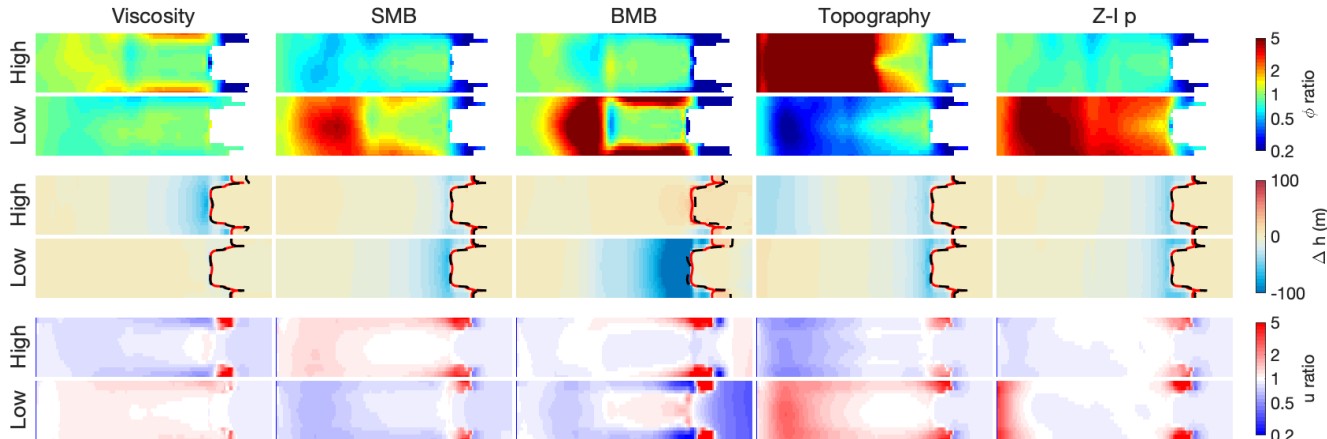

**Figure 7: Errors in inverted bed roughness, surface elevation, and surface velocity (relative to the target) for the perturbed inversions of experiment II. The top two rows show the errors in the bed roughness for the high and low perturbed inversions; the middle two**
**rows show the errors in the steady-state surface elevation; and the bottom two rows show the errors in the surface velocity. Each column represents a single perturbed model parameter: viscosity (i.e. Glen's flow law factor A), surface mass balance, basal mass balance, subglacial topography, and the Zoet-Iverson sliding law exponent $p$. The grounding line in the target (inverted) geometry is indicated by a solid red (dashed black) line.**

Finally, we perform a perturbed inversion for experiment II where we chose a non-equilibrated target geometry. We achieve
this by terminating the initialisation after 10,000 yr, instead of the default of 50,000 yr, so that the ice has only reached about 90% of its steady-state thickness. This non-steady-state geometry serves as the target for the inversion. Since the present-day observed geometry of the Antarctic ice sheet likely does not represent a steady state, but already displays sustained and accelerating thinning rates (Rignot et al., 2019), this experiment mimics the effects of erroneously assuming that the ice sheet is in equilibrium (a common assumption in modelling studies; Seroussi et al., 2019). The results of this experiment are shown
in Fig. 8. Here too, the inversion procedure results in very small errors in the ice geometry, relatively small errors in the velocity (note that the high velocity ratios occur in the slow-moving interior; in the fast-moving part of the ice stream, the errors are around 25%), but substantial errors in the bed roughness.

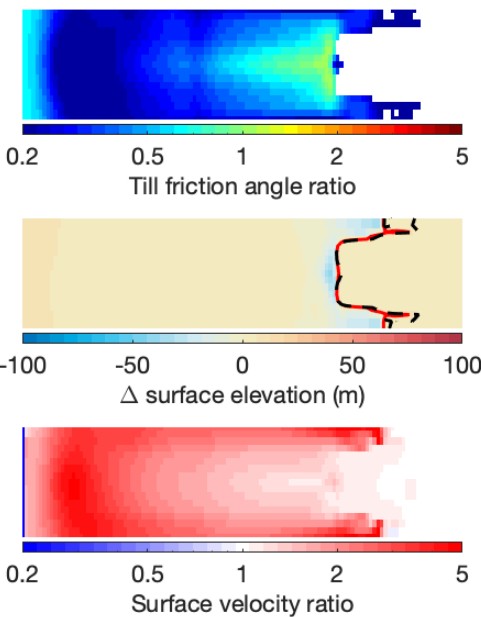

**Figure 8: Errors in inverted bed roughness (top), surface elevation (middle), and surface velocity (bottom) for the non-equilibrium**
**target inversion. The grounding line in the target (inverted) geometry is indicated by a solid red (dashed black) line.**

## 4.3 Dynamic ice-sheet response

To investigate the effect of compensating errors in basal inversions on the dynamic response of the ice sheet, we perform a
series of simulations based on experiment II, where we increase the basal melt, forcing the ice sheet to retreat. We use the
schematic basal melt parameterisation from the MISMIP+ Ice1r experiment (Asay-Davis et al., 2016) and run the model for
500 years. We initialise our simulations with the perturbed parameters, inverted bed roughness, and steady-state ice geometry
from the perturbed inversions presented in Sect. 4.2. For the "non-equilibrated" experiment, note that the ice sheet at the end
of the inversion is in a steady state; it has achieved this by lowering the bed roughness far enough to match the target geometry,
which was not in a steady state. The resulting ice volume above flotation (relative to the steady state at t = 0) and the mid-
stream grounding-line position over time for all experiments are shown in Fig. 9.

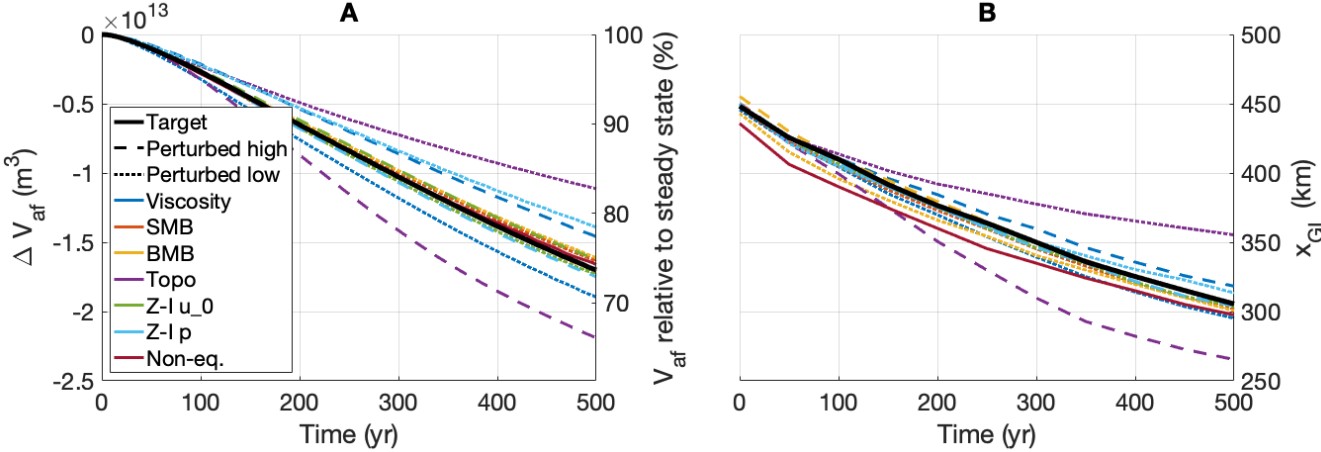


**Figure 9, panel A: change in ice volume above flotation ($\Delta V_{af}$), shown in absolute terms, as well as relative to the initial, steady-tate volume. Panel B: mid-stream grounding-line position ($x_{GL}$) over time in the perturbed retreat simulations of experiment II. Colours indicate the perturbed parameter; line styles indicate the direction of perturbation. The unperturbed simulation is shown by the solid black line.**

In the 500-year unperturbed simulation, the grounding line retreats by about 150 km, causing the ice volume above flotation

to decrease by about $1.7 \times 10^{13}$ m³. As a result of the introduced errors in the perturbed simulations, this mass loss is increased

(decreased) by up to 30% (35%) relative to the unperturbed simulation. The errors in the subglacial topography have the

strongest effect, with the high-perturbed run showing nearly twice as much ice loss as the low-perturbed run. This is followed

by the sliding law exponent (-18% to +3%) and the ice viscosity (-14% to +11%). The effects of the errors in the SMB, the

BMB, the sliding law transition velocity, and the non-equilibrated target geometry are small.

**5 Discussion**

We investigated the effects of compensating errors in basal inversions. We presented a novel geometry/velocity-based

inversion procedure, which produces good results in schematic experiments with a moving ice margin and grounding line, and

which produces robust convergence behaviour under an evolving ice geometry. We applied this method to two different

idealised-geometry experiments, where we quantified the errors in the inverted bed roughness that arise from perturbations in

other model parameters, such as the ice viscosity, mass balance, sliding law, and subglacial topography. We find that relatively

small perturbations in these parameters, which are generally within the uncertainty ranges for the Greenland and Antarctic ice

sheets, can lead to substantial compensating errors in the bed roughness. In our idealised experiments, these errors were often

larger than the actual spatial variations in bed roughness. This implies that one should be cautious in interpreting the outcome

of a basal inversion as an accurate physical representation of bed roughness underneath an ice sheet. We find that the dynamic

response of the ice to a retreat forcing is most sensitive to errors in the subglacial topography, followed by the ice viscosity

and the sliding law. Errors in the surface and basal mass balance appear to have only a small effect on the retreat, although this effect might become more pronounced when local instead of ice-sheet-wide errors are taken into account.

The aim of basal inversion procedures in many ice-sheet models is not to provide an accurate approximation of the actual bed roughness, but rather to produce an ice-sheet that matches the observed state in terms of geometry and/or velocity. The underlying assumption is that any compensating errors in the inverted bed roughness and other model components in terms of the ice geometry, will also compensate each other in terms of their effect on the ice sheet's dynamic response. We tested this assumption by using a basal inversion to initialise a number of different simulated ice sheets, all with slightly different model

parameters (viscosity, mass balance, etc.). We find that, even though the inversion results in all models have nearly identical steady state geometries, their dynamic response (represented here by the ice volume loss after a short period of forced ice-sheet retreat) can differ by as much as a factor of two. The strongest effect arises from the uncertainty in the subglacial topography, followed by the sliding law exponent and the ice viscosity. Uncertainties in the surface and basal mass balance lead to considerable errors in the bed roughness, but have only a small impact on the dynamic response, as does erroneously

assuming that the target (i.e. observed) ice-sheet geometry represents a steady state.

    The geometry of the experiment used to produce these findings describes a marine setting typical of West Antarctica, where the rate of mass loss under a forced retreat is mainly governed by ice-dynamical processes such as viscous flow and basal sliding (Seroussi et al., 2020). In a land-based setting more typical of the Greenland ice sheet, where most mass is lost through

atmospheric processes (Goelzer et al., 2020), the effects of these ice-dynamical uncertainties will likely be smaller. However, as long-term projections of sea-level rise under strong warming scenarios are dominated by marine-grounded ice loss in West Antarctica (Seroussi et al., 2020), such projections will likely contain substantial uncertainties as a result of the processes we described, possibly as large as 35% of the projected ice loss.

## 6 Conclusions

We have investigated the effect of compensating errors when deriving basal conditions underneath an ice sheet using inversion techniques. We find that errors in the modelled estimates of other physical quantities, such as the viscosity or subglacial topography of the ice, can substantially affect the estimated basal conditions. Our results imply that, even when basal inversion is used to achieve a stable ice sheet with the desired geometry, uncertainties in other model parameters can have a substantial effect on that ice sheet's dynamic response. Improving our knowledge of the ice sheet interior (temperature, rheology,

viscosity) and substrate (geometry, roughness) therefore should remain important goals of the glaciological community.

## Appendix A

In order to illustrate the convergence of our flowline-based inversion procedure, we performed additional simulations of the unperturbed versions of experiments I and II, where the inversion was allowed to run for 200,000 years. For comparison, we also ran the same simulations with the CISM-based inversion procedure. In this procedure, the rate of change $\frac{d\varphi}{dt}$ of the bed

roughness $\varphi$ is calculated based only on the local mismatch in the ice thickness $H$ and the surface velocity $u$:

$$\frac{d\varphi}{dt} = \frac{-\varphi}{\tau_c}\left(\frac{H_m - H_t}{H_0} - \frac{|u|_m - |u|_t}{u_0}\right). \tag{A1}$$

The values of the scaling parameters are $H_0 = 100$ m and $u_0 = 10$ m/yr. The timescale of adjustment $\tau_c$ is 10,000 years in

experiment I, and 40,000 years in experiment II. These values were determined experimentally as the lowest value (i.e., fastest convergence) that did not result in numerical instability. The results of experiment I are shown in Fig A1. Panel A shows the time evolution of the root-mean-square of the relative surface elevation mismatch $\frac{H_m - H_t}{H_t}$, the relative surface velocity mismatch $\frac{|u|_m - |u|_t}{|u|_t}$, and the bed roughness mismatch $\frac{\varphi_m - \varphi_t}{\varphi_t}$. These quantities converge to a stable solution that is typically within a few percent of the target, with the flowline-averaged approach presented in this study achieving smaller errors than the local-

mismatch approach from CISM. The fact that there is no overfitting can be seen in panel B, which shows the root-mean-square of the rate of change $\frac{d\varphi}{dt}$ of the bed roughness $\varphi$, which exponentially decays. Without proper regularisation, small-wavelength terms in the bed roughness solution can continue to increase in amplitude as the model is run forward; the effect of these terms on the velocity solution displays diminishing returns, so that bigger and bigger changes to the solution are needed to reduce the velocity/geometry misfit. This shows up in the convergence plot by a bed roughness rate of change that soon starts to

exponentially increase. The Gaussian filter-based regularization term in our approach prevents this type of overfitting from occurring.

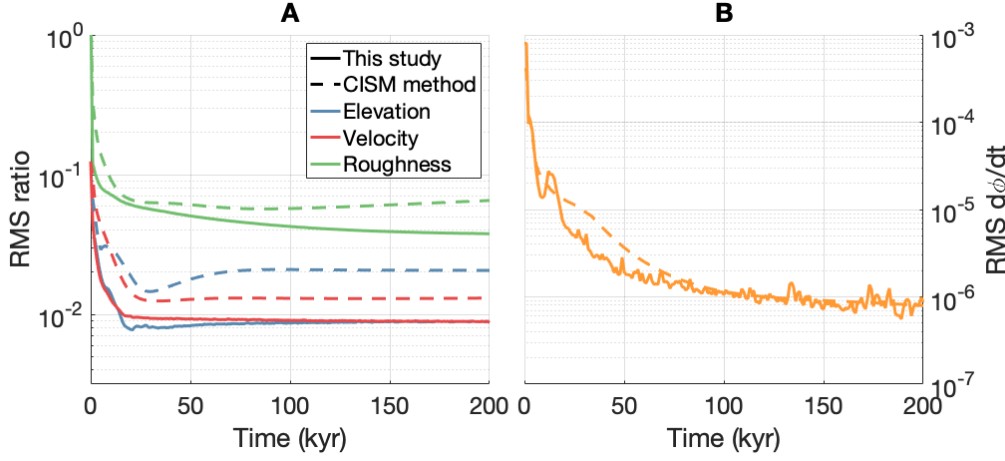

Fig. A2 shows the same quantities for experiment II. The sudden jump in the CISM-method results around 95,000 years is due to an advance of the grounding line by a single grid cell. We believe the wave-like features seen in the curve for the CISM-based approach in panel B, arise from an under-damped, slow oscillation between the bed roughness and the ice geometry. In the upstream part of the ice stream, where velocities are very low, the ice thickness responds very slowly to a change in bed roughness. Since the initial guess for the roughness there is too high, the ice starts to slowly accumulate; the inversion will respond by decreasing the roughness, but since the ice thickness changes very slowly, the roughness is reduced too much, causing the ice to eventually become too thin, etc. With the current choice of timescale of 40,000 yr, these oscillations eventually dissipate. Including a dH/dt-term in the inversion removes this problem; the velocity term in our own approach has a similar effect, since velocities respond instantaneously to a change in bed roughness.

The curve for our own inversion approach in Fig. A2, panel B, displays noise-like features. We believe these to be caused by an interaction between the velocity term in the inversion, the iterative solvers used in the stress balance solver (both for the linearised problem, i.e. with fixed effective viscosity, and for the non-linear viscosity iteration; see Berends et al., 2022), and the dynamic time step used for the ice thickness equation. The combination of these iterative solvers with a dynamic time step causes (very) small errors to continuously appear in the velocity solution, only to be repressed by the subsequently reduced model time step. For the fast-flowing ice of this particular geometry, these velocity errors start to affect the bed roughness inversion before they are repressed by the dynamic time step, which causes the "noise" that is visible in the curve of our approach in Fig. A2, panel B. Using smaller tolerances in the stop criteria for the two iterative solvers in the stress balance solver reduces this problem, at the expense of increasing the model's computational cost. Since Fig. A2, panel A shows that the resulting errors in the roughness solution do not accumulate, we deem this to be acceptable.

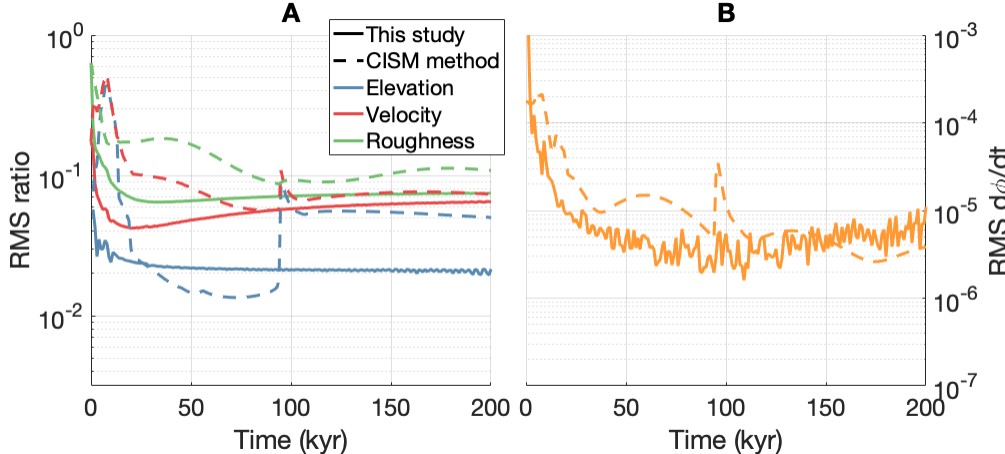

**Fig. A2: convergence of the inversion procedure for experiment II. Panel A: root-mean-square of the relative mismatches in surface elevation (blue), surface velocity (red), and bed roughness (green) over time, for both the flowline-averaged method presented here (solid lines) and the local-mismatch approach from CISM (dashed lines). Panel B: root-mean-square of the bed roughness rate of change, for both methods.**

*Acknowledgements*. We would like to thank Jorge Bernales and Willem Jan van den Berg for providing helpful comments during the execution of this project, as well as two anonymous reviewers for their helpful comments on the manuscript.

*Author contributions*. CJB performed the experiments and analysed the data. CJB wrote the draft of the manuscript; all authors contributed to the final version.

*Code and data availability*. The source code of IMAU-ICE, scripts for compiling and running the model on a variety of computer systems, and the configuration files for all simulations presented here, are freely available on Github: https://github.com/IMAU-paleo/IMAU-ICE.

*Competing interests*. The authors declare that they have no competing interests.

*Financial support*. CJB was supported by PROTECT. This project has received funding from the European Union's Horizon 2020 research and innovation programme under grant agreement no. 869304 (PROTECT; [PROTECT article number will be assigned upon acceptance for publication!]). TvdA was supported by the Netherlands Polar Program. The use of supercomputer facilities was sponsored by NWO Exact and Natural Sciences. Model runs were performed on the Dutch National Supercomputer Snellius. we would like to acknowledge SurfSARA Computing and Networking Services for their support. WHL was supported by the National Center for Atmospheric Research, which is a major facility sponsored by the National Science Foundation under Cooperative Agreement No. 1852977.

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
