# Peer review of "Compensating errors in inversions for subglacial bed roughness: same steady state, different dynamic response"

_The Cryosphere, 2022_

## Author Response (AR1)

**Response to comment tc-2022-103-RC1 by Anonymous Referee #1**

We thank the anonymous reviewer for their helpful comments on our manuscript, and would hereby like to address the concerns they raised. Reviewer comments are shown in italics, our responses in regular type.

**Major comments**

There are some confusions or inaccuracies at times about the two inversion methods. This is especially the case when referring to inversions overall and discussing the question of steady-state: while the steady-state assumption is used for the inversion based on transient simulations, this is not needed for "snapshot" inversions based on variation data assimilation. The geometry and velocity observations are used in the stress balance equation only, and the ice sheet does not need, nor is assumed, to be in steady-state. The combination of the given velocity and geometry will prescribe whether the ice sheet is in steady-state or evolves over time. Such terms are mostly used in the abstract and introduction and should be clarified to remove any possible ambiguity.

While there are nowadays transient inversion methods in use that do not make the steadystate assumption (e.g., in CISM a new approach is currently being tested that inverts for geometry, velocity, and thinning rates), we agree that most of the time this is still the case. We will clarify the distinction between the two families of inversion methods in the manuscript.

The results proposed show the impact of errors in various ice sheet fields on the inferred basal friction values in a relatively comprehensive way, but does not put this work in the context of previous studied. Previous work on inferring several fields (Arthern et al., 2015; Gudmundsson and Raymond, 2008), impact of rheology (Seroussi et al., 2013), or role of errors in observations (Habermann et al., 2012) has been done in the past and should be referenced and discussed to better describe the improvements and new results proposed here.

We agree that these are relevant references that should be mentioned in the manuscript. We will expand the introduction section of the manuscript to provide a more comprehensive overview of previous work on inversion methods, and specifically the effects of errors in model parameters and observations.

There is limited discussion on the choices made to perturb the different fields (viscosity, surface mass balance, etc.) and I would be curious to understand how these choices were made. Also, how do these changes compare to each other between the different fields (are these large or small bias) and how do they compare to our knowledge of the different fields and the current uncertainty for each of them? Such information would help better inform the results, and in particular the uncertainty in future evolution (Fig.9). The chosen parameter ranges are based both on upper-bound estimates of the uncertainty in observations and parameters for the Antarctic and Greenland ice sheets, and on the desire to stay within a range where the errors can be compensated by changing the bed roughness.

For Glen's flow law factor, we assume that this depends on the englacial temperature through an Arrhenius relation. The uncertainty in the surface temperature through the last glacial cycle is about ± 1 K for Antarctica (Jouzel et al., 2007), and about ± 4 K for Greenland (Alley et al., 2000; Kindler et al., 2014). Furthermore, in realistic applications a flow enhancement factor is often applied to account for anisotropic rheology and damage. Estimates for the values of these enhancement factors differ significantly (Ma et al., 2010). Based on the literature, an uncertainty of an order of magnitude in the ice flow factor seems plausible. We chose a smaller range (increase/decrease of 25%) to ensure that the inversion procedure was still able to reproduce the target geometry.

For the surface mass balance, we chose an admittedly small range of  $\pm$  5% in the accumulation rate. This is because, for simplicity's sake, we only changed the uniform accumulation rate. While observations of local accumulation rates, melt rates, and SMB contain significantly larger uncertainties, caused by both measurement errors/uncertainties and interannual variability, the integrated mass balance of the entire Greenland / Antarctic ice sheets is better constrained through e.g. GRACE observations. Based on the Greenland Surface Mass Balance Model Intercomparison Project (GrSMBMIP; Fettweis et al., 2020), a range of  $\pm$  5% for the long-term mean integrated mass balance seems plausible.

For the basal mass balance (or rather the sub-shelf melt rate, as we apply no melt underneath grounded ice), uncertainties are much larger (e.g. Burgard et al., 2022). Here too, we applied the simplest possible option of a uniform melt rate. As with the flow factor, our choice of parameter range was constrained not so much by observations, but by the need to stay within the window where the inversion procedure is able to reproduce the target geometry.

For the subglacial topography, we chose a range of  $\pm$  10% of the ice thickness. While the surface elevation of the Greenland and Antarctic ice sheets is generally known very accurately, estimates of ice thickness and bedrock elevation are based on interpolation of local radar measurements. In the BedMachine Greenland v4 dataset (Morlighem et al, 2017), the reported uncertainty in the bedrock elevation exceeds 10% of the ice thickness over about 30% of the ice sheet. Our choice of increasing/decreasing the estimated ice thickness by 10% everywhere serves as an upper bound, as it is unlikely that all of the data and extrapolations are biased in the same direction.

For the sliding law parameters, we chose a parameter range based on values reported in the literature. Zoet and Iverson (2020) report a range of transition velocities between 50 and 200 m/yr, whereas in CISM a default value of 200 m/yr is used. For the exponent, Zoet and Iverson report a value of 5, CISM uses a value of 3, and using a value of 1 makes it a linear sliding law (which is still used in some ice-sheet models), so testing a range of 1 to 5 covers all those options.

We will include this information in the manuscript.

Finally, there is an improvement described for the inversion technique, but this impact of this improvement on the inferred bed roughness and misfit with observed field is not shown. It would be interesting to add an experiment to compare the results with and without this improvement (e.g., using the non-steady-state case).

The inversion technique described here improves upon the one recently added to CISM. That one uses the mismatch between the modelled and target surface elevation and surface velocity, but does not have the flowline-averaging procedure. The difference is mostly noticeable in the stability, the speed of convergence, and in the reduction of artefacts at icemargin or grounding-line grid cells. The final inverted roughness fields are nearly identical (apart from those artefacts), which is logical as the inclusion of a proper regularisation term guarantees that there exists a unique answer to the inverse problem. To illustrate the difference in performance, we will add a short Appendix to the manuscript, where we show the convergence of the modelled surface elevation, surface velocity, and inverted bed roughness, for both the CISM method and our new flowline-averaged method. This shows that both methods converge to a stable solution (so no overfitting), but the flowlineaveraged method does so significantly faster.

Technical comments

p.1 l.22: "retreat" -> "mass loss"

We will change this.

p.1 I.26: the ABUMIP experiments are not only idealized cases but also extreme experiments remove all the ice shelves around Antarctica. These results are therefore showing an extreme case, and are difficult to compare with experiments using more nuanced forcing. It would be good to nuanced this paragraph.

We will clarify that this is an extreme case.

p.2 I.30: maybe not just basal sliding and roughness but basal conditions overall.

We will change this.

*p.2 l.34: this paper demonstrated the role of the basal sliding law used, but did not really conclude on the bed roughness itself.*

We will clarify this.

p.2 I.53: as explained here, this second method is performed using observations at a given time, and the model is not run forward in time as part of the inversion procedure. The geometry is therefore "given" but not really "kept fixed" as there is no notion of time.

We will reflect this conceptual difference between "nudging" vs. data assimilation methods here and throughout the manuscript.

p.2 I.54: it would be important to mention how this method works: a cost function, measuring the distance between some observed fields and their modeled equivalent, is defined and this cost function is minimized during the inversion procedure.

We will include this short technical description in the manuscript.

p.3 I.65: you need to make a distinction between the two methods here: what is described only works for the first inversion technique, in the second one, there is no impact of the ice sheet geometry as part of the inversion and therefore no "steady-state" ice sheet. This should be rephrased to either distinguish the two methods or to make the description generic enough to cover both methods. (same with "thinning the ice" on I. 66)

We will clarify that these compensating errors will affect both families of inversion procedures in similar but different ways.

p.3 I.78: remove "still"

We will do so.

p.3 l.82: remove "of" (of as a result)

We will do so.

p.3 I.75-83: this is an accurate description of what is done in the paper, maybe the abstract is a bit too generic, which can lead to confusions about the two overall methods to infer properties (variational data assimilation and adjustment during a long transient run)

We believe that the two families of inversion procedures are fundamentally similar enough (particularly since ours also includes a velocity term, and is generally well able to match the observed velocities even when compensating errors are present) that the more general conclusions stated in the abstract are justified.

p.4 l.101: add some references in this paragraph (sliding laws, etc.)

We will include references to Weertman (1957) for the power-law sliding, and to Iverson et al. (1998) for Coulomb sliding.

p.5 l.135: explain what the "I" variables are. Also, why use the entire flowlines and not a region of influence with a given region of influence around the various points/regions?

The "I" variables represent the normalised half-flowline integrals of the ice thickness and velocity errors. The scaling functions wu and wd ensure that errors along the flowline close to p receive more weight than those further away. We did not experiment with limiting the length of flowline to integrate over. The length of the region influenced by changes in the bed roughness might well be variable, so that using the entire flowline seemed like a simpler first choice. We will include this information in the manuscript.

**p.5 Eq.6-7: It's not clear if/how these variables defined for a single point are extended to the entire domain. Are the I variables defined and used locally or globally?**

All equations are evaluated (i.e. a flowline traced in both directions, the integrals calculated, and the rate of change of the roughness calculated) for every ice-covered grid cell individually. We will mention this in the manuscript.

**p.6 l.155: Can you detail these artefacts and the impact of the scaling values used?**

The artefacts present as individual or clustered grid cells (typically on the ice margin or grounding line) where the iterative bed roughness adjustment overshoots, quickly diverging to extreme values. Also, sometimes short-wavelength oscillations in the roughness field can occur when the roughness is adjusted too fast relative to the amount of applied regularisation. The scaling values serve to limit the rate of adjustment so that these divergences do not occur. We will clarify this in the manuscript.

*p.6 l.160:* \tau was used for the basal shear stress so it might better to use a different letter for the time scale.

We will replace \tau with t\_s.

**p.6 l.171: How do these terms compare to the regularization terms used in other methods?**

In velocity-based inversions using more complex mathematical tools to minimise an explicitly defined cost function, typically a term related to the gradient or curvature of the roughness solution is included in the cost function, which supresses small-wavelength terms in the solution. Pattyn (2017) applies a Savitzky–Golay filter during the nudging process, similar to our Gaussian filter; Pollard and DeConto (2012) do not report any regularisation term. In CISM, no regularisation is applied, although the inclusion of a dH/dt term in the calculation of dphi/dt likely results in some smoothing. We will mention this in the manuscript.

**p.7 l.191: target geometry and velocity**

We will change this.

p.9 l.217: "a deep oceanic trough" -> "a deep ocean" (there is no trough on the ocean part on Figure 3)

We will change this.

p.9 l.221: What value is used for A?

The flow factor A = 1.13928E-17 Pa-3 yr-1 is tuned to obtain a steady-state mid-channel grounding-line position at x = 450 km. We will state this in the manuscript.

Sections 3.1 and 3.2: How do you grow these two configurations to a steady-state and how long does it take to grow them? What is the resolution of the model?

A steady state is achieved in both experiments by running the model forward for 50,000 years. For experiment 1, we use resolutions of 40, 20, and 10 km (only 20 km for the perturbed inversions). For experiment 2, we use 5 and 2 km (only 5 km for the perturbed inversions and the retreat simulations). We will state this in the manuscript.

p.10 l.232: What is the impact of using a different initial value for \phi? 5 degrees is the most common value in this set-up so it might be good to make sure this initial value does not influence the results of the inversion!

Choosing a different initial value generally does not impact the final inverted roughness field. However, choosing a value that is far away from the target means it can take longer to converge to the correct answer, as the ice-sheet geometry will initially adjust to the "wrong" roughness, and will take some time to relax back to the correct geometry. We will state this in the manuscript.

p.10 Fig.5: It would be good to change the colorbars for the surface elevation and surface velocity differences and better see the errors. In caption, change "ice-sheet geometry" to "ice-sheet surface elevation". Finally, the colorbars for Fig. 4 and 5 are the same, but the velocity is very different for the two simulations, so it would make sense to adjust the values and really focus on the velocity modeled for each experiment.

We will change the colormap for the velocity errors to make it clear that these panels show a different quantity than the ones showing the till friction angle error. We will change the colour limits to show the errors more clearly. We will change the caption to mention surface elevation instead of geometry. Lastly, we will update the figures, as the caption stated that the relative velocity errors were shown but the figures themselves accidentally showed the absolute errors. We will apply these changes to all figures.

We do not think that using a different scale for the target velocity fields in Figs. 4 and 5 (the unperturbed experiments) is desirable. The narrow ice stream on the southern margin of the ice sheet in experiment I (Fig. 4) reaches a maximum velocity of over 700 m/yr (visible as the narrow purple strip in the figure), which is not that much slower than the ~1000 m/yr reached by the shelf in experiment II.

p.11 Fig.5: In caption, change "ice-sheet geometry" to "ice-sheet surface elevation" and "three unperturbed" to "two unperturbed". Same as Fig. 4 for the colorbars of the surface elevation and surface velocity differences.

We will do so.

p.11 I.251: Mention this is for experiment 1.

The perturbations listed here apply to both experiments. We will clarify this.

p.11 I.259: How about the role of local errors due to noise in observations? For example a random noise of 5 or 10% in the velocity, thickness, etc.? How long is the model run to reach the steady-state? And how do you know that this steady-state is reached?

The inversions are initialised with the target geometry, and are run for 100,000 years. Generally, the inverted bed roughness converges to a stable solution within ~30,000 years, so that this is more than long enough to reach a steady state. We will state this in the manuscript. The figures in the new Appendix also clearly show this.

We have not performed any experiments concerning observational errors, apart from the "topo" experiments that represent a systematically over/underestimated ice thickness.

p.12 Fig.6: again I would focus the colorbar for the surface elevation difference given the relatively small range of errors. Caption: "steady-state geometry" -> "steadystate geometry and velocity". Add "from left to right" before the list of parameters studies (viscosity, surface mass balance, ...)

We will do so, and apply the changes to the colourmaps and colour limits mentioned earlier.

*p.13 l.284: How do you define that these are acceptable? How do they compare to observational errors?*

The errors in the inverted surface elevation and velocity here are undoubtedly larger than observational errors. We deem them acceptable because they are well within the range of errors for initialised ice models in intercomparisons such as initMIP-Greenland, etc. We will mention this in the manuscript.

*p.13 l.292: Over what part of the domain do you perturb the basal mass balance (grounded, floating, both)?*

We apply only a sub-shelf melt rate. We will clarify this in the manuscript.

p.13 l.294: "introduced errors" -> "errors introduced"

We will change this.

*p.13 l.299: "except for the inversions with perturbed basal melt rate" (or rephrase to make that more clear)*

We will change this.

p.13 l.299: missing word after "these"

We will fix this.

p.14 l.313: For which experiment?

For experiment II. We will state this in the manuscript.

p.14 l.315: I am not sure to interpret that correctly: is the inversion or the run providing the data for the inversion run only to 10% of the steady-state? The second case would better represent reality, though as the ice sheet is never in steady-state, neither case is ideal.

The target run providing the data for the inversion is run to about 90 % of the steady-state ice thickness. We will clarify this in the manuscript

p.14 l.318: These errors are actually not so large compared to some of the runs with perturbed fields and the errors are located only upstream of the grounding line.

That is correct.

p.15 I.326: remove "steady-state" (last case is not steady-state)

The ice sheet from the inversion run is in steady state; the target for the last inversion is not. We will clarify this in the manuscript.

p.15 l.328: indicate where along the y axis is this grounding line shown?

The time-series show the mid-stream grounding-line position. We will clarify this in the manuscript.

*p.*15 Fig.9: It would be good to add the percentage of mass loss on the right of panel A.

We will do so.

p.16 l.338: remove "negligibly"

We will do so.

*p.16 l.340: Change tense in paragraph (We investigated instead of We have investigated, etc.)*

We will do so.

p.16 I.340: How does that compare to the previous approach? It would be good to show one case with the previous and new approach to see the impact of the changes (maybe the non steady-state MISMIP+).

This is explored in the newly added Appendix.

p.16 l.346: this is not really new

We agree, but since we still come across people who place a lot of trust in their inverted roughness fields, repeating this conclusion seems useful.

p.16 I.356: How do the different changes applied in the different fields compare to our knowledge (lack of knowledge) in these fields? What are the implications for comparing these uncertainties?

We believe that our choice of parameter ranges generally reflects the uncertainty of these quantities for the Greenland and Antarctic ice sheets (see also our earlier response regarding these choices). We will mention this in the manuscript.

p.16 l.361: "not clear": what did you try?

We did not perform any additional experiments. Our use of the phrase "discrepancy" here might be confusing, as it is maybe not surprising that not all physical processes affect both basal sliding and ice-dynamical response identically. We will remove this sentence.

p.17 l.363-369: need more references and justifications

We will add more references to substantiate the claims made in this paragraph.

**Response to comment tc-2022-103-RC2 by Anonymous Referee #2**

We thank the anonymous reviewer for their helpful comments on our manuscript, and would hereby like to address the concerns they raised. Reviewer comments are shown in italics, our responses in regular type.

**Major comments**

The literature review is not adequate and at times not accurate. In the introduction (lines 46-57) the authors mention several papers as examples of bed roughness inversion, where in fact most of those papers target the inversion of the basal drag (or basal friction), not the bed roughness. While these quantities can be related, they are certainly not interchangeable. Also I think there are some relevant papers that should be cited. Babaniyi et al, TC 2021, present a rigorous approach on how to account for model errors (in particular in the rheology) when inverting the basal friction. Other studies that look at the impact of rheology on inverted quantities are Seroussi et al., Journal of Glaciology 2013 and Ranganathan, Journal of Glaciology, 2020. A preliminary study of how errors in SMB could affect inverted basal parameters where featured in perego et al, JGR, 2014.

We agree that these are relevant references that should be mentioned in the manuscript. We will expand the introduction section of the manuscript to provide a more comprehensive overview of previous work on inversion methods, and specifically the effects of errors in model parameters and observations. We will also take care to clarify the difference in inverting for bed roughness, and inverting directly for basal drag.

The authors present a clever but involved and ad-hoc way to invert for the bed roughness. I find this anachronistic. Nowdays, the large majority of work performing inversion of ice sheet quantities uses PDE-constrained optimization approaches, which are very well understood and naturally linked to Bayesian inference problems.

Variations on the "nudging" method of inversion are used in e.g. CISM (Lipscomb et al., 2021), PISM (Albrecht et al., 2020), and f.ETISh (Pattyn, 2017), which are some of the most widely-used ice-sheet models of today.

Key parts of the PDE-constrained optimization problem are the regularization terms, that avoid overfitting, and the ability to weigh observations according to their trustworthiness (i.e. root mean square errors in observations). The proposed method has a regularization step in the form of a Gaussian filtering, but it's not clear to me how to link that to the typical regularization term in the formal optimization approach. In my understanding, their method does not account for root means square errors in the velocity or thickness data, which is a significant limitation. I think the author should discuss these limitations and also investigate how different choices of the radius of the Gaussian filters affect their results. I suspect that there is too much overfitting in their inversion. Regarding regularisation and overfitting: our inversion simulations are ran for 100,000 years. Typically, the inverted bed roughness converges to a stable solution within the first 50,000 years. This means that the regularisation (which indeed is done by way of a simple Gaussian smoothing) is working well, preventing the inversion from continuing to adapt the roughness solution when the misfit is no longer significantly reduced. Furthermore, Figs. 4 and 5 clearly show that no visible small-wavelength terms appear in the roughness solution, again indicating that there is no significant overfitting occurring. To illustrate this further, we will add a short Appendix that shows the errors over time in the modelled surface elevation, surface velocity, and inverted bed roughness, as well as the rate of change of the inverted bed roughness. The errors all converge to a stable, non-zero value, while the rate of change of the bed roughness exponentially decays. All of this indicates that no overfitting occurs.

The radii of the two Gaussian filters in our approach, were arrived at during preliminary experiments. The values reported here are the lowest values we found that effectively repressed small-wavelength overfitting terms in the inverted bed roughness. The target roughness in our experiments is relatively "smooth", with horizontal variations on a scale that is at least an order of magnitude larger than the grid resolution. Increasing the radii of the filters did not affect our inverted solution much until it was increased to several grid cells, so that it approached the spatial scale of the roughness variations. Roughness variations of a smaller spatial scale could therefore potentially be obscured by the smoothing in our approach. However, these would then quickly approach the ice-dynamical limit of roughness variations that can be resolved by inverting from surface observations (about 50 ice thicknesses; Gudmundsson and Raymond, 2008), so we do not believe that this would pose a serious problem in practical applications. We will add this information to in the manuscript.

Our method currently does not include weighting of the velocity/elevation mismatch based on uncertainty estimations in the observations. It would not be difficult to include these weights in the method, and it is certainly something worth considering when we move on to apply this method to the Greenland or Antarctic ice sheet. For the idealised experiments we present here, there is of course no observational error. We will add these thoughts to the manuscript.

**Minor comments**

eq. (1): In general,  $\tau b$  and ub are vectors. Please write the equation in vector form (using the vector ub and its magnitude |ub|).

We will fix this.

eq. (1): How do you compute N?

In the experiments presented here, we set the effective overburden pressure N equal to the ice overburden pressure, assuming no subglacial water anywhere. We will mention this in the manuscript.

line 172: how do you choose the radii of the Gaussian filters?

See our earlier response regarding regularisation.

section 4.2. Typically we distinguish errors in the data (e.g. in velocity/thickness observation and, possibly, SMB), from model errors (specific laws and model parameters like A, p, etc). The latter are harder to account for. I think it would be better to do this distinction in your perturbed experiments.

We agree that this distinction is important to make. We will clarify this in the manuscript.

Figs 4 and 4: The range of the colorbar for the bed roughness is too wide. I would limit it to the interval [0.5,2] or so, rather than [0.1,10]. More ticks on the colorbar might help as well.

Based also on the comments of reviewer #1, we will change Figs. 4 - 8 to have a smaller range for the colour scales for all three errors (roughness, elevation, and velocity). We will also change the colour map for the velocity, to make it more clear that these panels show a different quantity than the roughness. Also, the velocity errors were accidentally still shown as absolute errors rather than relative; we will fix this.

---

## Author Response (AR2)

**Response to comment tc-2022-103-RC1 by Anonymous Referee #1**

We thank the anonymous reviewer for their helpful comments on our manuscript, and would hereby like to address the concerns they raised. Reviewer comments are shown in italics, our responses in regular type.

> *Line 163: In general, the presence of regularization terms does not guarantee the uniqueness of the minimum. Moreover, local minima might be present so that different convergence paths might lead to different local minima. Please explain or rephrase the sentence.*

We will rephrase this sentence.

> *Line 543: I don't understand why the fact that the inversion method converges implies that the method does not overfit. Please explain, maybe add a reference, or remove the sentence.*

Without a regularization term, short-wavelength terms in the solution can continue to increase in amplitude as the model is run forward; the effect of these terms on the velocity solution displays diminishing returns, so that bigger and bigger changes in the solutions are needed to reduce the velocity/geometry misfit. This shows up in the convergence plot by a bed roughness rate of change that soon starts to exponentially increase (up to a certain point; in our model code, the till friction angle phi is limited between 0.1 and 50 degrees, so at some point the solution "stabilizes" by reaching theses limits everywhere). The Gaussian filter-based regularization term in our approach prevents this type of overfitting from occurring. We will state this in the manuscript.

> *Fig A2, right: can you explain why both curves of the bed roughness rate of change seem to be increasing towards the end of the simulation? Are these simulation not converged yet?*

For the CISM approach, we believe the wave-like features arise from an under-damped, slow oscillation between the bed roughness and the ice geometry. In the upstream part of the ice stream, where velocities are very low, the ice thickness responds very slowly to a change in bed roughness. The initial guess for the roughness is too high, causing the ice to slowly accumulate; the inversion will start lowering the roughness, but since the ice thickness changes very slowly, it lowers the roughness too much, causing the ice to eventually become too thin. With the current choice of timescale (tau = 40,000 yr) these oscillations do eventually dissipate, but it takes a long time. Including a dH/dt-term in the calculation of dphi/dt removes this problem, which is what Bill and Tim now use in their newest inversion approach. In our own approach, the velocity term in dphi/dt has a similar effect, since velocities respond instantaneously to a change in bed roughness (in theory).

For our own inversion procedure, we believe the noise-like features in the dphi/dt-curve to be caused by an interaction between the velocity term in our inversion, the iterative solvers

used in our stress balance solver (both for the linearised problem, i.e. with fixed effective viscosity, and for the non-linear viscosity), and the dynamic time step used for the ice thickness equation. The combination of these iterative solvers with a dynamic time step causes (very) small errors to continuously appear in the velocity solution, only to be repressed by the subsequently reduced model time step. For the fast-flowing ice of this particular geometry, these velocity errors start to affect the bed roughness inversion before they are repressed by the dynamic time step, which causes the "noise" that is visible in the dphi/dt-curve of our approach. Using smaller tolerances in the stop criteria for the two iterative solvers in our stress balance solver reduces this problem, at the expense of increasing the model's computational cost. Since the left panel of the figure shows that the resulting errors in the roughness solution do not accumulate, we deem this to be acceptable.

We will state this in the manuscript.

**Response to comment tc-2022-103-RC1 by Anonymous Referee #1**

We thank the anonymous reviewer for their helpful comments on our manuscript, and would hereby like to address the concerns they raised. Reviewer comments are shown in italics, our responses in regular type.

**Overall comments**

> *The results of the different experiments made in the paper are described and discussed one by one in section 4, and there is limited discussion comparing the different results together and addressing some overall questions. Two important aspects that should be discussed in more details: 1) the problem of overfitting mentioned by a previous reviewer that should be included in the main text*

The problem of overfitting is explored in detail in Appendix A, which is now extended based on the new comments of Reviewer #1 to discuss how Fig. A2, panel B shows that overfitting is not problematic with our method. We will additionally refer to this Appendix again in Sect. 2.2 where overfitting is mentioned.

> *and 2) the question of which parameters cause the most problems based on the perturbation experiments done in the study, and the ones we can relatively safely ignore, based on the values used in the manuscript and the current uncertainties in these fields.*

We will add a few lines to the first paragraph of the Conclusions, discussing the relative importance of uncertainties in the different parameters.

**Technical comments**

> *l. 55-60: could use more references*

We will additionally refer to Athern and Gudmundsson (2010), Gagliardini et al. (2013), and Arthern et al. (2015) as examples of studies that have used velocity-only-based inversion methods to estimate basal slipperiness or traction.

> *l. 139: the exponent is p =*

The current sentence is grammatically correct.

> *l. 164: has a unique solution*

This sentence has been replaced based on the comments by Reviewer #1.

> *l. 194: $I_2$ and $I_3$ are reversed in the description compared to the equations*

They are not.

*l.211: These values are*

We will change this.

*l.320: It is stated that the initial conditions do not impact the results. It would be good to provide some numbers for that.*

The conclusion that choosing a different initial uniform value for the bed roughness does not affect the final inverted roughness field, was based on preliminary experiments, which we unfortunately did not store.

*l.325: "very small" should be quantified*

The errors in the unperturbed experiments are typically < 5% for the bed roughness, < 5 m for the surface elevation, and < 5% for the surface velocity. We will state these numbers in the manuscript.

*l.445-450: Additional references needed*

We will refer to Rignot et al. (2019) to support the claim that the present-day Antarctic ice sheet is not in equilibrium, and to Seroussi et al. (2019) to support the claim that many ice-sheet models used for future projections (implicitly) assume that it is.

---

## Author Response (AR3)

**Response to comment tc-2022-103-RC1 by Anonymous Referee #1**

We thank the anonymous reviewer for their helpful comments on our manuscript, and would hereby like to address the concerns they raised. Reviewer comments are shown in italics, our responses in regular type.

> *The latest version of the manuscript "Compensating errors in inversions for subglacial bed roughness: same steady state, different dynamic response" by C. J. Berends et al. addresses most of the comments raised by both reviewers. There are a just couple of points that remain unclear to me.*

> *The first one is the answer to the question of the initial conditions impact mentioning simulations have been lost. I understand that could be expensive to rerun but the phrasing of the answer is a little surprising and it might be best to remove this entirely or to write it differently.*

We will remove the reference to the preliminary experiments that demonstrated the (lack of) effect of the choice of initial bed roughness. We will also add a reference to Appendix A, where we show that the inversion converges within about 30,000 years.

> *The second one is the answer to the question of inverse method convergence and overfit. Yes the presence of regularization is needed, but I don't see the link with the different parts of the explanations in the answer to this question, so this might need a bit more clarification.*

We will add another reference to Appendix A in the paragraph on regularization in the methodology section (lines 200 – 215). Hopefully this will clarify to the reader how our regularization approach works, and how the experiments presented in the Appendix demonstrate the absence of significant overfitting.